# The Physico-Chemical Properties of Glipizide: New Findings

**DOI:** 10.3390/molecules26113142

**Published:** 2021-05-24

**Authors:** Giovanna Bruni, Ines Ghione, Vittorio Berbenni, Andrea Cardini, Doretta Capsoni, Alessandro Girella, Chiara Milanese, Amedeo Marini

**Affiliations:** 1C.S.G.I.—Department of Chemistry, Physical-Chemistry Section, University of Pavia, Via Taramelli 16, 27100 Pavia, Italy; ines.ghione01@universitadipavia.it (I.G.); vittorio.berbenni@unipv.it (V.B.); doretta.capsoni@unipv.it (D.C.); alessandro.girella@unipv.it (A.G.); chiara.milanese@unipv.it (C.M.); amedeo.marini@unipv.it (A.M.); 2A.M.S.A. Anonima Materie Sintetiche Affini S.p.A., Viale Giuseppe Di Vittorio 6, 2100 Como, Italy; a.cardini@amsacomo.it

**Keywords:** glipizide, differential scanning calorimetry, thermogravimetric analysis, decomposition, polymorph, 5-methyl-*N*-[2-(4-sulphamoylphenyl)ethyl]pyrazine-2-carboxyamide, kinetic study, shelf-life

## Abstract

The present work is a concrete example of how physico-chemical studies, if performed in depth, are crucial to understand the behavior of pharmaceutical solids and constitute a solid basis for the control of the reproducibility of the industrial batches. In particular, a deep study of the thermal behavior of glipizide, a hypoglycemic drug, was carried out with the aim of clarifying whether the recognition of its polymorphic forms can really be done on the basis of the endothermic peak that the literature studies attribute to the melting of the compound. A number of analytical techniques were used: thermal techniques (DSC, TGA), X-ray powder diffraction (XRPD), FT-IR spectroscopy and scanning electron microscopy (SEM). Great attention was paid to the experimental design and to the interpretation of the combined results obtained by all these techniques. We proved that the attribution of the endothermic peak shown by glipizide to its melting was actually wrong. The DSC peak is no doubt triggered by a decomposition process that involves gas evolution (cyclohexanamine and carbon dioxide) and formation of 5-methyl-*N*-[2-(4-sulphamoylphenyl) ethyl] pyrazine-2-carboxamide, which remains as decomposition residue. Thermal treatments properly designed and the combined use of DSC with FT-IR and XRPD led to identifying a new polymorphic form of 5-methyl-*N*-[2-(4-sulphamoylphenyl) ethyl] pyrazine-2-carboxamide, which is obtained by crystallization from the melt. Hence, our results put into evidence that the check of the polymorphic form of glipizide cannot be based on the temperature values of the DSC peak, since such a peak is due to a decomposition process whose Tonset value is strongly affected by the particle size. Kinetic studies of the decomposition process show the high stability of solid glipizide at room temperature.

## 1. Introduction

Glipizide (Figure 1), *N*-(4-[*N*-(cyclohexylcarbamoyl)sulfamoyl] phenethyl)-5-methylpyrazine-2-carboxyamide (*GPZ*), a second-generation sulfonylurea derivative, is an oral hypoglycemic drug used to treat type II diabetes. Its therapeutic use is due to the stimulation of the release of insulin through interaction with the SUR1 receptor on K/ATP-sensitive channels in β cells membrane [1,2,3]. Furthermore, it can increase insulin sensitivity and decrease hepatic glucose production by an indirect extra pancreatic mechanism [4]. It is completely absorbed when administered orally but, having a short half-life, is rapidly metabolized in the liver, and its inactive metabolites are excreted in urine [5,6].

*GPZ* is very slightly soluble in water and in an acidic environment. R. Matsuda et al. reported a solubility of 20 mg L^−1^ in water at room temperature [7] and similar results, under the same conditions, were obtained by N. Seedher and M. Kanojia, whose study also revealed a higher solubility in basic rather than acidic conditions [8]. Due to its very low solubility in water, glipizide is classified as class II by BCS (biopharmaceutical classification system); therefore, its bioavailability is limited by the dissolution rate [9].

According to the studies published in the scientific literature, *GPZ* melts at a temperature higher than 200 °C, and different melting temperatures have been reported [10,11,12]. The crystal structure of glipizide has been recently addressed by single-crystal X-ray diffraction data [13]. It crystallizes in the triclinic *P-1* Space Group, Z = 2, *a* = 5.1578(3) Å, *b* = 8.9760(5) Å, *c* = 23.9704(13) Å, a = 83.247(2)°, b = 85.543(2)°, g = 78.987(2)°, V = 1080.01(11) Å^3^ at 100 K. The crystal structure data are deposited in the Cambridge Crystallographic Data Centre (CCDC N. 1516974). In the last few years studies have been performed with the aim to identify new polymorphs of this compound [14,15,16]. This interest arises from the need to improve its dissolution profile in order to achieve a higher bioavailability [17,18]. Renuka et al. [14] claimed the identification of six different crystalline forms (forms I, II, III, IV, V, and VI; all obtained by different crystallization methods) on the basis of a different thermal behavior. However, both the experimental design of their measurements and the conclusions drawn from these raised some doubts. Our main criticisms are related to the following issues: (1) the differential scanning calorimetry (DSC) measurements were performed at a single heating rate; (2) the DSC peaks were integrated with non-uniform criteria; (3) an inappropriate comparison was made between XRPD (X-ray powder diffraction) patterns; (4) no TGA (thermogravimetric analysis) measurements were performed, which can provide important information about thermal stability; (5) the deduction of the monotropic relationship between the polymorph forms was based on weak considerations.

Another way to improve the pharmaceutical behavior of poorly soluble drugs is to use them in the amorphous form, which, by virtue of its higher free energy, shows higher solubility and dissolution rate [19,20]. The amorphous form can be obtained by grinding, but to avoid its intrinsic tendency to recrystallize, a proper selection of the process parameters is necessary [19,20,21,22,23]. K. Xu et al. [15,16] defined the process parameters to ball-mill glipizide form I, identified by Renuka et al. [14], that allow its transition to form II (amorphous form) and form III. However, also in this work, the experimental set up seems to be inappropriate, and the conclusions drawn from the experimental evidence are not convincing. Indeed, more attention and a deeper analysis should be paid to the mass changes measured in TGA.

In the present work the physico-chemical process underlying the endothermic peak present in the DSC curve of glipizide, generally attributed to the melting of the active principle, was deeply investigated. Since its temperature could be used to evaluate the polymorphic purity of industrial batches of glipizide, it is important to verify the reliability of this control method.

A wide range of analytical techniques was used: DSC, TGA, simultaneous DSC–TGA (SDT), XRPD, Fourier-transform infrared spectroscopy (FT-IR), and scanning electron microscopy (SEM). In addition, great attention was paid to the experimental set up and to the complementary interpretation of the results obtained by all the techniques.

## 2. Materials and Methods

### 2.1. Materials

Four production batches and two standard USP batches (USP R09020, USP H1H396) of *GPZ*, together with a batch of one of its synthesis intermediates, 5-methyl-*N*-[2-(4-sulphamoylphenyl) ethyl] pyrazine-2-carboxyamide (*GS* in the following), were kindly provided by AMSA S.p.A. (Como, Italy).

### 2.2. Ground Samples

One of the production batches was ground by a planetary mill (Planetary Micro MillPulverisette 7, Fritsch) in an agate ball-milling jar containing 50 balls of 5 mm in diameter of the same material, with a 500-rpm rotation speed. Grinding periods of 10 min were alternated with intervals of 5 min to avoid an overheating of the powder. The powder was ground for 120 (*GPZ2h*), 180 (*GPZ3h*), 300 (*GPZ5h*), and 480 min (*GPZ8h*).

### 2.3. Analytical Techniques

#### 2.3.1. Thermal Measurements

Simultaneous DSC–TGA measurements were performed by a SDT Q600 interfaced with a TA5000 data station (TA Instruments, New Castle, DE, USA). The measurements were performed in open standard aluminum pans under two different atmospheres, nitrogen flow (50 mL·min^−1^) or static air, at several heating rates (β): 0.5, 1, 2, 5, 10, and 20·K·min^−1^.

Conventional DSC and TGA measurements were carried out by a DSC Q2000 apparatus and a TGA Q5000 apparatus, both interfaced with a TA 5000 data station (TA Instruments, New Castle, DE, USA). The DSC instrument was calibrated using ultrapure (99.999%) indium (m.p. = 156.6 °C; ΔH = 28.54·J·g^−1^) as standard. The heating rates used were: 0.3, 0.5, 1, 2, 5, 10, 20, and 30 K⋅min^−1^. The measurements were performed on sample amounts of about 3–4 mg in open pans under nitrogen flow (25 mL⋅min^−1^) or in air. Furthermore, both cyclic measurements with different final temperatures in the first heating and measurements with an isothermal step were performed.

#### 2.3.2. FT-IR Measurements

For the FT-IR analysis a Nicolet FT-IR iS10 spectrometer (Nicolet, Madison, WI, USA) equipped with ATR (attenuated total reflectance) sampling accessory (Smart iTR with diamond plate) was used. Thirty-two scans in the 4000–600 cm^−1^ range at 4 cm^−1^ resolution were coadded.

#### 2.3.3. XRPD

XRPD measurements were performed by using a Bruker D5005 diffractometer (Bruker BioSpin, Fällanden, Switzerland) using CuKα radiation (λ = 1.54056 Å), graphite monochromator, and scintillation detector. The patterns were collected in air at room temperature in step scan mode with a step size of 0.02° and counting time of 2 s per step in the angular range 2θ of 5–35 ° (voltage = 40 kV, current intensity = 40 mA).

#### 2.3.4. SEM

SEM images were collected with a Zeiss microscope Evo MA10 (Carl Zeiss, Oberkochen, Germany). Before the analysis, the samples were sputtered with carbon to make them conductive.

## 3. Results and Discussion

### 3.1. Preliminary Studies

Appendix A, Appendix A, shows the DSC curves, all collected at 10 K·min^−1^ from different batches. The shape of the endothermic peak (asymmetrical) is common to all batches, but its onset temperature (*T_onse_*_t_) can be appreciably different. In particular, all the production batches by AMSA S.p.A. (trace a of Appendix A is representative of all production batches) and the USP H1H396 batch (trace b of Appendix A) provided identical results (within the experimental error). The *T_onset_* was 214.5 ± 0.8 °C and the enthalpy change (Δ*H*) was 290.8 ± 0.3 J·g^−1^.

The thermal behavior of the USP R09020 batch was instead somewhat different (trace c). While the enthalpy change was the same obtained for the USP H1H396 batch, the *T*_onset_ of the peak (204.9 ± 0.8 °C) was significantly lower.

Additionally, for the *GPZ3h* sample (trace d) the enthalpy change was identical (within the experimental error) to those already discussed, while the peak *T_onset_* (206.3 °C ± 0.8 °C) was lower than that of the original production batches (and then of USP H1H396), but similar to that of USP R09020.

To summarize, except for the USP R09020 batch, all the batches analyzed as received gave identical calorimetric results (*T*_onset_ and Δ*H*) within the experimental error. The milling process (sample *GPZ3h*) lowered the *T*_onset_ of the peak without modifying the enthalpy change.

USP R09020 differed from the other batches only for the appreciably lower *T*_onset_ of the peak.

The fact that the measured enthalpy was identical for all the batches examined suggests that they all have the same chemical/phase composition. This is confirmed by the perfect superimposition of the XRPD patterns and of the FT-IR spectra collected on the samples of the different batches (Appendix A). The XRPD pattern well is comparable to the simulated one, obtained by the crystal structure reported in literature [13]: the investigated samples display the triclinic crystal structure. The reason why we observe different *T*_onset_ for the USP R09020 batch will be discussed further below once additional knowledge has been gained about the process behind the DSC peak.

In the following, the study of the thermal behavior of glipizide will be deepened. Unless otherwise specified, the measurements were performed on just one of the production batches.

### 3.2. Simultaneous DSC–TGA Measurements

#### 3.2.1. Simultaneous DSC–TGA Measurements under Nitrogen Flow

As can be seen in Figure 2, the DSC endothermic peak is associated with a significant mass loss measured on the TG curve. Furthermore, the measurements carried out at heating rates ranging from 1 K·min^−1^ to 20 K·min^−1^ show that the endothermic peak shifts towards gradually increasing temperatures with increasing β. The shift concerns both *T*_onset_ and *T*_f_ (see Table 1; *T*_f_ is the final temperature of the peak), but if the strong dependency of *T*_f_ on β is expected, it is, on the contrary, a significant variation of *T*_onset_, which should be substantially independent of the heating rate in a melting process, is unexpected.

Another noteworthy aspect is that the mass loss associated with the peak (also reported in Table 1) was constant at any heating rate. A mass loss suggests that a decomposition process takes place, and this could also explain the observed *T*_onset_ dependency on β. However, the fact that the mass loss remained constant while occurring in different temperature ranges suggests that the underlying process is not a simple evaporation/sublimation of a chemical substance but a real transformation that leads to a chemical entity of definite composition, different from the initial one. The mass change was −27.7% ± 0.7%, and it was measured by analyzing in each case the derivative thermogravimetric curve (DTG curve). As it is well known, the determination of the enthalpy change requires the exact integration of the peak, which is difficult in this case because of the quite different levels of the baseline before and after the peak. Furthermore, the Δ*H* values measured with the SDT instrument are less quantitatively reliable than those measured with a DSC instrument. These are the reasons why the curves obtained with the simultaneous instrument were not used for the measurement of enthalpy changes.

#### 3.2.2. Simultaneous DSC–TGA Measurements in Air Atmosphere

The thermal behavior of glipizide in an air atmosphere was completely similar to that observed under nitrogen flow. The DSC peak and the associated mass change were, indeed, in the identical temperature range previously observed. Moreover, the mass change in air—also independent of the heating rate—was −27.8% ± 0.5%, i.e., identical, within the experimental error, to that obtained in nitrogen atmosphere. These findings indicate that the decomposition underlying the DSC peak is not an oxidative process, since it proceeds to an identical extent in air and in nitrogen atmosphere.

### 3.3. DSC and TGA Measurements

In order to confirm the framework deduced by SDT measurements and to perform an accurate quantitative analysis, the thermal measurements were also performed by independent DSC and TGA apparatuses. Heating rates from 30 K·min^−1^ to 0.5 K·min^−1^ were used. The *T*_onset_ of the endothermic peak ranged from 217.7 ± 0.4 °C at 30 K·min^−1^ to 196.1 ± 0.1 °C at 0.5 K·min^−1^ (Table 2 and Appendix A, Appendix A). The peak enthalpy change remained constant at 291.0 ± 0.6 J·g^−1^.

TGA measurements confirmed that the mass loss associated with the calorimetric peak was independent of β and agreed, within the experimental error, with that measured by the simultaneous apparatus. In addition, the TG curves obtained in air perfectly superimposed to those under nitrogen flow (Appendix A, Appendix A), and the effect of increasing β was merely to shift the entire TG curve to higher temperatures (Appendix A, Appendix A). Finally, the mass loss was the same in nitrogen (−27.9% ± 0.2%) and in air atmosphere (−27.8% ± 0.2%), which confirmed the non-oxidative nature of the decomposition process.

#### 3.3.1. Comments on SDT, DSC, and TGA Measurements

Our deductions from thermal measurements can be summarized as follows:What underlies the DSC peak is not a melting process but a decomposition process. This is proven by two experimental findings: (a) its *T*_onset_ significantly increases with increasing β, and this behavior is not compatible with a melting process; (b) the DSC peak is associated with a consistent mass loss.The decomposition is non-oxidative, as proven by the fact that the mass loss is the same both in air and nitrogen atmospheres.

Therefore, the attribution of the glipizide DSC peak to a simple melting process is not correct [10,11,12]. The calorimetric peak is undoubtedly triggered by a decomposition process, which involves a gaseous product evolution and then a mass loss under the peak. The fact that such a mass loss is constant with changing β (and then the temperature range in which the decomposition takes place) suggests that, in all cases, the same gaseous product is released by the sample and the same decomposition product is obtained. It is worth analyzing this aspect, verifying if, starting from the amount of the mass loss, the nature of the gaseous phase released by the sample and then the nature of the decomposition residue can be deduced.

Furthermore, it is worth noting that the experimental evidence described so far does not require—but neither does it exclude—the co-presence of a melting process under the DSC peak. This aspect also deserves clarification by further measurements able to determine whether the decomposition is accompanied or followed by the melting of the solid residue.

#### 3.3.2. The Nature of the Decomposition Reaction

Assuming that the solid compound obtained by decomposition is in 1:1 stoichiometric ratio with *GPZ*, from *m* grams of glipizide, corresponding to $\frac{m\left(g\right)}{445.54\left(g\cdot {\mathrm{mol}}^{-1}\right)}=m\cdot 2.244\cdot 10^{-3}$ moles, we should obtain an identical number of moles of decomposition residue. Since the decomposition involves a mass loss of 27.9% (mean value obtained from TG measurements under nitrogen flow), the mass (grams) of the decomposition residue must correspond to $m\cdot \frac{100-27.9}{100}$. Given that both moles and mass values of the decomposition residue are known, it is easy to calculate the molar mass of the solid residue: $\frac{m\cdot \frac{100-27.9}{100}\left(g\right)}{m\cdot 2.244\times 10^{-3}\left(\mathrm{mol}\right)}=321.2\;\left(g\times {\mathrm{mol}}^{-1}\right)$. This value is very similar to the molar mass of 5-methyl-*N*-[2-(4-sulphamoylphenyl) ethyl] pyrazine-2-carboxamide (320.4 g·mol^−1^), which represents the compound that could remain as residue, assuming that the decomposition takes place with the breaking of the amide bond colored in red in the molecular structure reported in Figure 3, according to the A reactive scheme in Figure 4. In this case the mass change would be caused by the loss of cyclohexanamine (since its boiling temperature is 134 °C, it should be in vapor phase at the DSC peak temperature values) and carbon dioxide.

Then, the quantitative data support the hypothesis that the DSC peak corresponds to the decomposition of glipizide with the formation of 5-methyl-*N*-[2-(4-sulphamoylphenyl) ethyl] pyrazine-2-carboxamide, which remains as solid residue.

It is worth noting that the glipizide decomposition could also take place with the breaking of the amidic bond colored in green in Figure 3, which would lead to the formation of 1-cyclohexyl-3-[[4-(2aminoethyl) phenyl] sulfonyl] urea and 5-methyl-2-pyrazinecarboxylic acid, according to the B reactive scheme reported in Figure 4. However, we consider that this second pathway is unlikely since 5-methyl-2-pyrazinecarboxylic acid has a boiling temperature of 316 °C.

With the aim to confirm the hypothesized decomposition scheme, thermal measurements with isothermal steps were performed to obtain *GPZ* samples that had undergone a controlled decomposition in order to compare them with the *GS* samples, precisely constituted by 5-methyl-*N*-[2-(4-sulphamoylphenyl) ethyl] pyrazine-2-carboxamide. The comparison was carried out by DSC, FT-IR and XRPD measurements.

#### 3.3.3. Measurements with Isothermal Steps

The TGA measurements with isothermal steps at different temperatures and for different time ranges were performed according to Table 3. The selected temperature values were significantly lower than *T*_onset_ and even more than *T*_f_, which define the thermal range in which the decomposition takes place in dynamic conditions: the extent of the decomposition reaction depends on both temperature and time so that the use of isothermal temperatures lower than those of the dynamic measurements is compensated by the longer time available to complete the reactive process in the isothermal measurements. At each temperature the isothermal steps were allowed to proceed for at least the time necessary to reach a constant mass. Relatively low values of isothermal temperatures have the advantage of avoiding the melting process of the hypothesized decomposition residue (whose melting point is 235 °C). At the end of the isothermal steps, the samples, named as reported in Table 3, were analyzed by both FT-IR and XRPD.

FT-IR spectra of $GPZ_{170\;{}^{\circ}C}^{210\;min}$, $GPZ_{175\;{}^{\circ}C}^{150\;min}$, $GPZ_{180\;{}^{\circ}C}^{135\;min}$ and $GPZ_{185\;{}^{\circ}C}^{125\;min}$ samples were almost indistinguishable from each other and from the spectrum of the *GS* sample, but they definitely differed from that of *GPZ*. As an example, in Figure 5 the spectra of $GPZ_{170\;{}^{\circ}C}^{210\;min}$ and $GPZ_{185\;{}^{\circ}C}^{125\;min}$ are reported, representing the two samples treated at the two ends of the isothermal step temperatures.

The XRPD patterns of these samples are also very similar to that of the *GS* sample, as can be observed in Figure 6, where the results obtained for $GPZ_{170\;{}^{\circ}C}^{210\;\min}$ and $GPZ_{185\;{}^{\circ}C}^{125\;\min}$ samples are reported.

Then, the FT-IR and XRPD characterization of the samples obtained after the isothermal steps confirms the reaction scheme according to which glipizide, after heating, undergoes decomposition with formation of 5-methyl-*N*-[2-(4-sulphamoylphenyl) ethyl] pyrazine-2-carboxamide.

At this point, our scheme of what happens when heating glipizide is quite clear. Glipizide, when heated, decomposes. The decomposition produces a solid residue and a gas phase, which evolves from the sample and is responsible for the mass loss observed. The gas phase is cyclohexanamine, the solid phase is 5-methyl-*N*-[2-(4-sulphamoylphenyl) ethyl] pyrazine-2-carboxamide (*GS* as defined in Section 2.1), which is an intermediate of the glipizide synthesis and was provided by AMSA S.p.A., together with *GPZ* batches. The study of the thermal behavior of *GS* will allow us to deepen our knowledge about the decomposition of glipizide.

#### 3.3.4. Cyclic DSC Measurements on GS

Cyclic DSC measurements with heating up to 250 °C were performed on *GS* (Figure 7).

In the first heating, a melting endothermic peak at *T*_onset_ = 237.9 °C with Δ*H* = 161.2 J·g^−1^ was present. In the following cooling (2 K·min^−1^), an exothermic crystallization process was present at a temperature significantly lower than that of the melting process (*T*_onset_ = 174.4 °C and Δ*H* = −120.6 J·g^−1^). During the second heating, an endothermic peak with *T*_onset_ = 214.3 °C and Δ*H* = 134.4 J·g^−1^ appeared.

Reiterating the cycle several times, i.e., heating and cooling the sample repeatedly, the exothermic peak of the first cooling and the endothermic one of the second heating were reproduced both in temperature and in enthalpy change values. This suggests that, once melted, *GS* crystallizes in a polymorphic form *GS**, whose melting temperature and enthalpy are different from those of the original form and that, unlike this, is stable to melting since it can be melted and obtained again by crystallization of the melt. We noted that the shape of the crystallization peak of *GS** (i.e., of the polymorph that invariably forms from the melt) was rather strange (see the insert of Figure 7). The crystallization process was so fast that the heat released caused an increase in the temperature of the sample, despite the fact that the temperature of the calorimetric cell in which the sample was contained was decreasing at a rate of 2 K·min^−1^. This happens because the crystallization is considerably undercooled (174.4 °C vs. 214.3 °C of the subsequent melting) and, consequently, occurs as a substantially instantaneous process; it is then foreseeable that the significant self-heating of the sample during crystallization leads to an underestimation of the enthalpy of crystallization, whose absolute value is in fact lower than that of the melting enthalpy (120.6 J·g^−1^ vs. 134.4 J·g^−1^).

The evidence from DSC measurements was confirmed by FT-IR and XRPD techniques. The measurements were performed on the *GS* sample, which was melted up to 250 °C and then cooled at 2 K·min^−1^ to room temperature in DSC. The sample was coded *GS** because, according to our speculations, it should have been transformed, following the cooling after the melting, into the polymorph *GS**. The XRPD patterns and FT-IR spectra of *GS** are compared with those of *GS* in Figure 8 and Figure 9 respectively.

It is immediately apparent at first glance that the XRPD patterns of the two samples are quite dissimilar and reflect different crystallographic structures.

On the contrary, as expected, the differences between the FT-IR spectra of the two samples are less evident: this happens because the technique shows the absorption of functional groups characteristic of the molecules, and the functional groups remain unchanged in the different polymorphic forms of the compound. The different crystallographic structure of the two polymorphic modifications reflects a different relative arrangement of the molecules of the compound, against which more or less evident shifts of the absorption bands of the functional groups can only be expected. Indeed, Table 4 shows that the absorption peaks/bands of *GS* are all present and shifted to a lower or higher wavenumber in the *GS** sample, confirming that the two samples correspond to polymorphic modifications of the same compound.

To our best knowledge, in literature, the existence of polymorphs of *GS* (5-methyl-*N*-[2-(4-sulphamoylphenyl) ethyl] pyrazine-2-carboxamide) is unknown.

#### 3.3.5. Cyclic DSC Measurements on Glipizide

Now that the thermal behavior of *GS* (5-methyl-*N*-[2-(4-sulphamoylphenyl) ethyl] pyrazine-2-carboxamide) has been studied in some detail, there is a key question: why is the *GS* melting not observed in the DSC traces (Appendix A), if, according to our evidence, it represents the decomposition residue of glipizide? At all heating rates, the *T*_max_ values of the decomposition peak, in correspondence to which a large portion of the sample had decomposed, were lower than the melting temperature of *GS*, but the return of the peak to the baseline (i.e., the part following *T*_max_) did not show any discontinuity due to melting of the obtained *GS*. However, the return of the peak to the baseline was very slow, which could be justified by an endothermic process extended over a temperature range rather than by the melting of a crystalline phase. To deepen this point, two Glipizide samples were prepared. One, coded *GPZ_250_*, was obtained by heating a *GPZ* sample up to 250 °C and then cooling it (at 2 K·min^−1^) down to room temperature. The other one, coded $GPZ_{250}^{180}$, was a glipizide sample heated to 250 °C, then cooled (at 2 K·min^−1^) to room temperature, heated again up to 180 °C, and then cooled back to room temperature. The XRPD patterns of the two samples are reported in Figure 10, together with that of *GS** (which is the same pattern shown in Figure 8).

It seems clear that: (a) *GPZ_250_* is mainly amorphous; (b) heating this sample up to 180 °C ($GPZ_{250}^{180}$) causes an at least partial crystallization; (c) the diffraction peaks of $GPZ_{250}^{180}$ show a lower intensity but they have the same angular positions of those of the *GS** sample.

Therefore, these results show that: (a) the residue obtained from glipizide decomposition is a mainly amorphous form of *GS*; (b) this amorphous form crystallizes to the polymorphic modification *GS**.

We can explain the shape of the DSC peaks, especially the very slow return to the baseline, considering that, in the experimental conditions used for the DSC measurements, the glipizide decomposition produced amorphous *GS*: an amorphous phase does not melt at a defined temperature, but it softens in a wide temperature range by a prolonged endothermic process, exactly as it happened in our DSC curves.

#### 3.3.6. Obtaining Crystalline GS (5-methyl-*N*-[2-(4-sulphamoylphenyl) ethyl] pyrazine-2-carboxamide)

To complete the picture, we still must verify if it is possible to identify the experimental conditions of the DSC measurements that allow us to obtain *GS* in the crystalline form instead of the amorphous phase from the glipizide decomposition. Obviously, if a crystalline form of *GS* could be obtained, we could see its melting peak during the DSC scan. With this aim, it is necessary: (1) to lower the initial temperature of the glipizide decomposition as much as possible; (2) to restrict the thermal range necessary to complete the decomposition process so that its final temperature is below the melting point of the crystalline form of *GS* as much as possible; (3) to provide the amorphous phase, originated from the decomposition, enough time for the transition in the crystalline phase. The operating mode to pursue, at the same time, all the requirements described, is to further lower the heating rate, compared to the DSC measurements previously discussed. In Figure 11, the DSC curve recorded on *GPZ* at 0.3 K·min^−1^ is shown. The *T*_onset_ and *T*_f_ values of the decomposition peak were 193.8 ± 0.1 °C and 205.6 ± 0.2 °C, respectively. The decomposition peak was followed by an endothermic peak with *T*_onset_ = 238 °C, corresponding to the melting of the polymorphic modification *GS*. As expected, in this experiment *GS** was not obtained, because, as noted above, this polymorph can form only from the *GS* melt.

#### 3.3.7. Batches of Glipizide with Different DSC Peak *T*_onset_: Explanation

Before closing the discussion about the interpretation of the thermal measurements, it is important to return to the preliminary DSC measurements described in Section 3.1. It was seen that, except for the USP R09020 batch, all the numerous analyzed batches of glipizide provided identical results, within the experimental error, both for the *T*_onset_ and the Δ*H* values of the DSC peak. The USP R09020 batch distinguished itself from the others because of the lower *T*_onset_, but not for the Δ*H* value. However, as we observed in paragraph 3.1, the chemical composition of this batch was identical to that of all the other batches analyzed.

Figure 12 shows the SEM images of three batches of glipizide: a production batch, USP H1H396, and USP R09020. It was well evident that the mean size of the particles of the USP R09020 batch were significantly lower than those of the production batch and of the USP H1H396 batch. The lower *T*_onset_ of the DSC peak of USP R09020, the only characteristic that distinguished it from the other batches, was due to the lower particle size.

### 3.4. Kinetic Study of the Thermal Decomposition

The shift of the maximum temperature of the DSC peak (*T*_max_) when changing the heating rate can be considered for the determination of the kinetic parameters of the decomposition process. For this determination we adopted two methods frequently used for the kinetic study in solid phase by dynamic thermal measurements: Kissinger and Ozawa–Flynn–Wall methods [24,25,26,27].

#### 3.4.1. Kissinger Method

The data used to verify the linear relationship predicted by the Kissinger equation are reported in Appendix A, while in Appendix A, the experimental data points and their linear regression are reported. The equation obtained is:$$\ln\left(\frac{\beta }{T_{\max}^{2}}\right)\;=79.812-\frac{43,805}{T_{\max}}\;R^{2}=0.9928$$

The good correlation coefficient (R^2^) confirms the existence of a linear relationship between the variables.

From the slope of the straight line, the apparent activation energy of 364.2 kJ·mol^−1^ is obtained.

#### 3.4.2. Ozawa–Flynn–Wall Method

The data used to verify the linear relationship predicted by the Ozawa–Flynn–Wall equation [24,25,26,27] are reported in Appendix A, while in Appendix A, the data points and their linear regression are reported. The equation obtained is:$$\log\beta =-\frac{19,442}{T_{\max}}+40.894\;R^{2}=0.9931$$

In this case as well, a good correlation coefficient between variables is obtained.

From the slope of the straight line, the apparent activation energy of 353.9 kJ·mol^−1^ is obtained, which is close to the value obtained by the Kissinger method.

The pre-exponential factor A was determined by Equation (1) under the E698-16 standard method stated by the ASTM International American regulatory body [28].
(1)$$A=\beta \;\left(\frac{E_{a}}{{R\;T}_{\max}^{2}}\right)\;\exp\left(\frac{E_{a}}{{R\;T}_{\max}}\right)$$

This method suggests replacing β with an intermediate value between those used for β (and using the value measured at that β as *T*_max_). Here, 5 K·min^−1^ was used as β.

Using the E_a_ value deducted from the Kissinger method (364.2 kJ·mol^−1^), A = 1.8 × 10^39^ min^−1^ is obtained, while using the E_a_ deducted from the Ozawa–Flynn–Wall method (353.9 kJ·mol^−1^), A = 1.3 × 10^38^ min^−1^ is calculated.

Substituting these values of activation energy and pre-exponential factor in the Arrhenius equation (Equation (2)), the kinetic constants of the decomposition at 25 °C (298.15 K) were obtained (Table 5). Assuming that the fractional lifetime is independent of the initial concentration of the reagent, as it happens in first-order kinetic, Equations (3) and (4) were applied to determine half-life ($t_{½}$) and shelf-life (t_90_) values at 25 °C. The kinetic data obtained by the two methods (Kissinger and Ozawa–Flynn–Wall) are reported in Table 5. The results indicate that glipizide is extremely stable at room temperature and has a particularly long shelf-life, which is certainly longer than the conservation period required for pharmaceuticals.
(2)$$K=A\cdot \exp\;\left(-\frac{E_{a}}{R\;T}\right)$$
(3)$$t_{½}=\frac{0.693}{K}$$
(4)$$t_{90}=\frac{0.105}{K}$$

## 4. Conclusions

This work allowed us to obtain new information about the thermal behavior of glipizide, which can be of interest both from a scientific point of view and from a practical point of view, (i.e., for the pharmaceutical industry).

First, we shed light on the process responsible for the endothermic peak present in the DSC curves of glipizide, and the reason why different batches of this compound can show the above peak at a different temperature was revealed, correcting the literature data attributing the peak to a melting process whose temperature can be used to differentiate the polymorphic forms of the compound.

Thanks to a proper experimental design, we obtained strong evidence that the DSC peak corresponds to a decomposition process. It was proven that 5-methyl-*N*-[2-(4-sulphamoylphenyl) ethyl] pyrazine-2-carboxamide is obtained from the decomposition reaction, and several well-designed thermal treatments, in addition to the combined use of thermal techniques, FT-IR and XRPD, allowed us to find a new polymorph of this compound, unknown until now.

The kinetic studies of the decomposition process, performed with two different non-isothermal methods based on the shift of the maximum temperature of the DSC peak with the heating rate, show the high stability of the solid glipizide at room temperature.

## Figures and Tables

**Figure 1 molecules-26-03142-f001:**
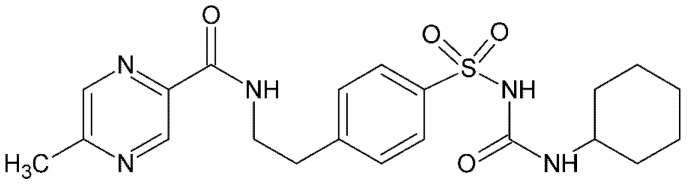
Glipizide molecular structure.

**Figure 2 molecules-26-03142-f002:**
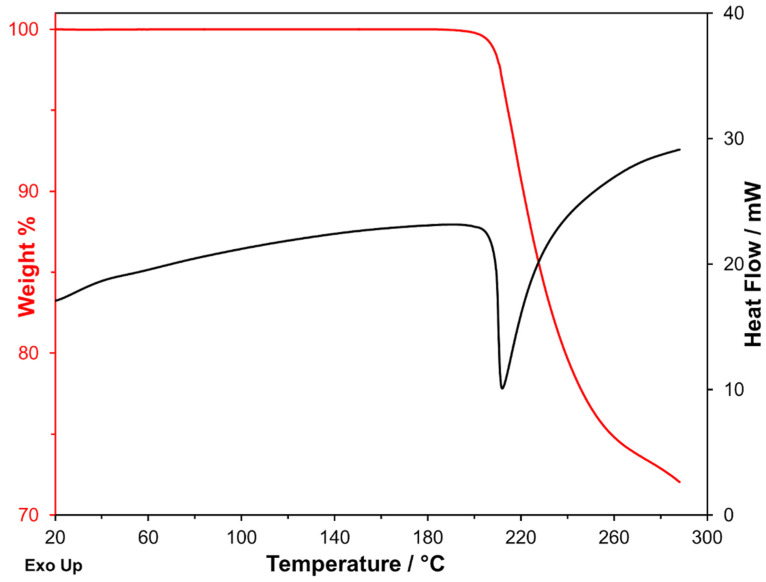
Simultaneous DSC–TGA curves recorded on one of the production batches at 10 K·min^−1^ under nitrogen flow.

**Figure 3 molecules-26-03142-f003:**
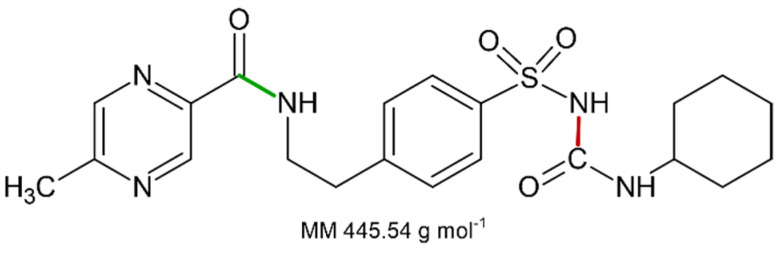
Molecular structure of *GPZ*. The two chemical bonds that could be subjected to breaking are colored in red and green.

**Figure 4 molecules-26-03142-f004:**
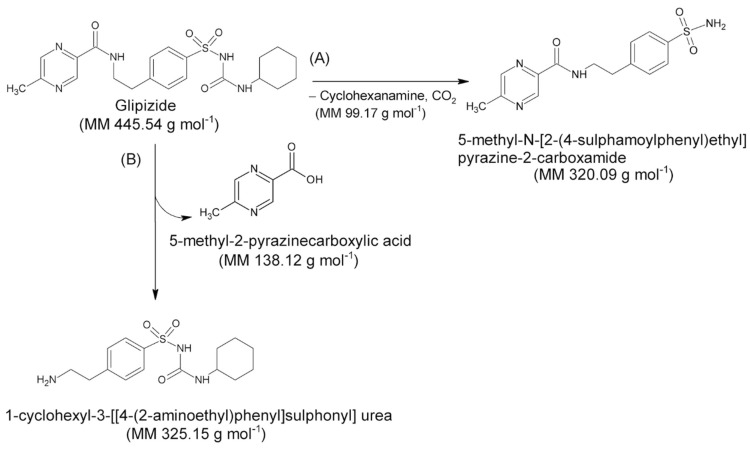
Decomposition reaction schemes hypothesized for the *GPZ* molecule. (**A**) Formation of 5-methyl-*N*-[2-(4-sulphamoylphenyl) ethyl] pyrazine-2-carboxamide; (**B**) Formation of 1-cyclohexyl-3-[[4-(2aminoethyl) phenyl] sulfonyl] urea.

**Figure 5 molecules-26-03142-f005:**
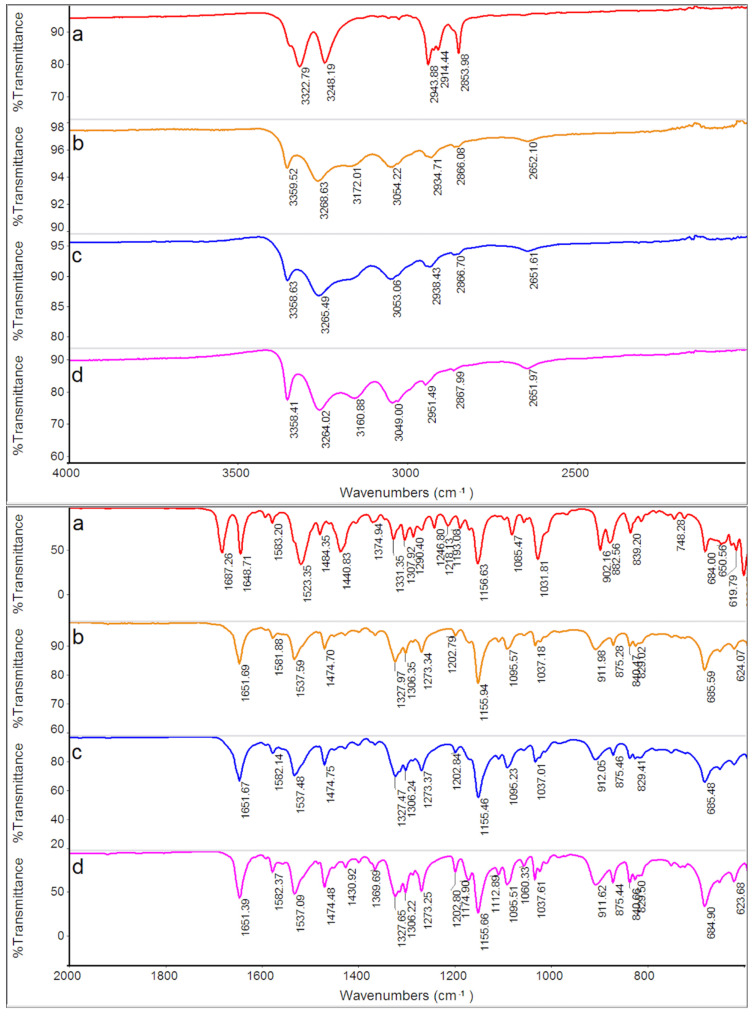
FT-IR spectra of a *GPZ* production batch (**a**), $GPZ_{170\;{}^{\circ}C}^{210\;\min}$ (**b**), $GPZ_{185\;{}^{\circ}C}^{125\;\min}$ (**c**), and *GS* (**d**) samples.

**Figure 6 molecules-26-03142-f006:**
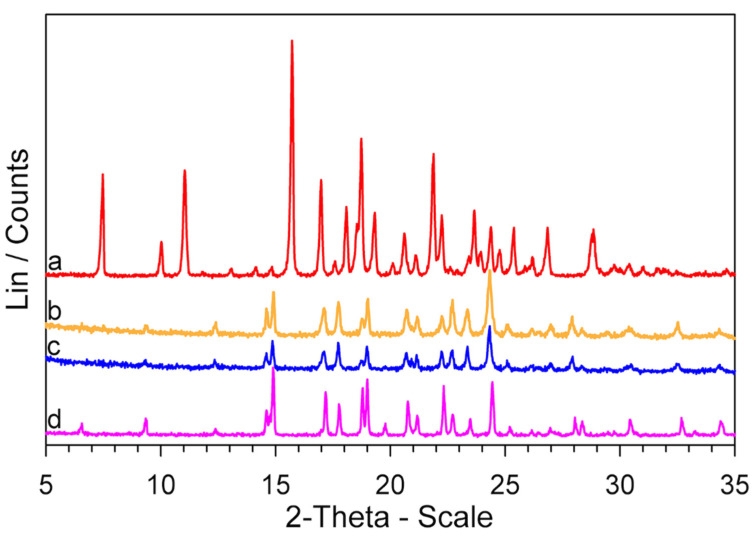
XRPD patterns of a *GPZ* production batch (**a**), $GPZ_{170\;{}^{\circ}C}^{210\;\min}$ (**b**), $GPZ_{185\;{}^{\circ}C}^{125\;\min}$ (**c**), and *GS* (**d**) samples.

**Figure 7 molecules-26-03142-f007:**
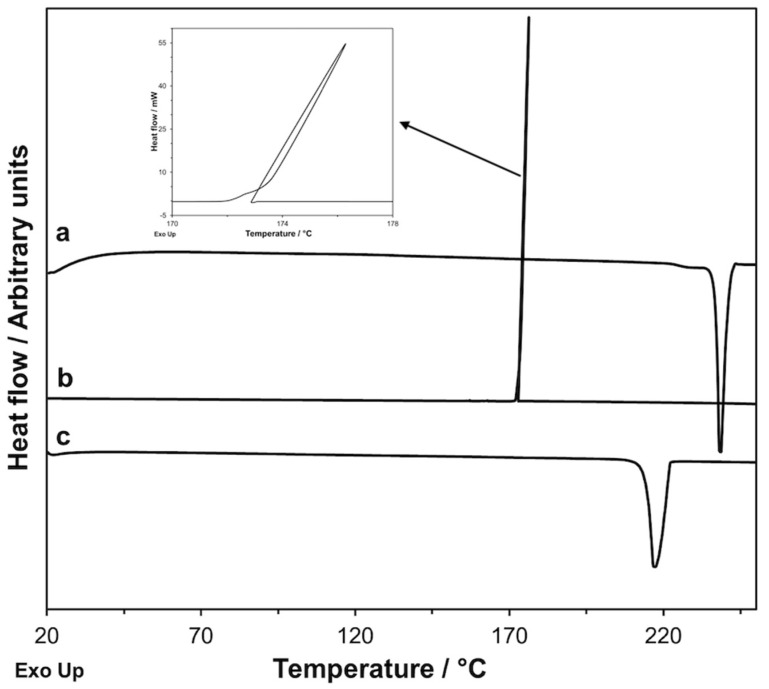
DSC curves recorded on a *GS* sample: first heating at 10 K·min^−1^ (**a**), cooling at 2 K·min^−1^ ((**b**) and insert), and second heating at 10 K·min^−1^ (**c**).

**Figure 8 molecules-26-03142-f008:**
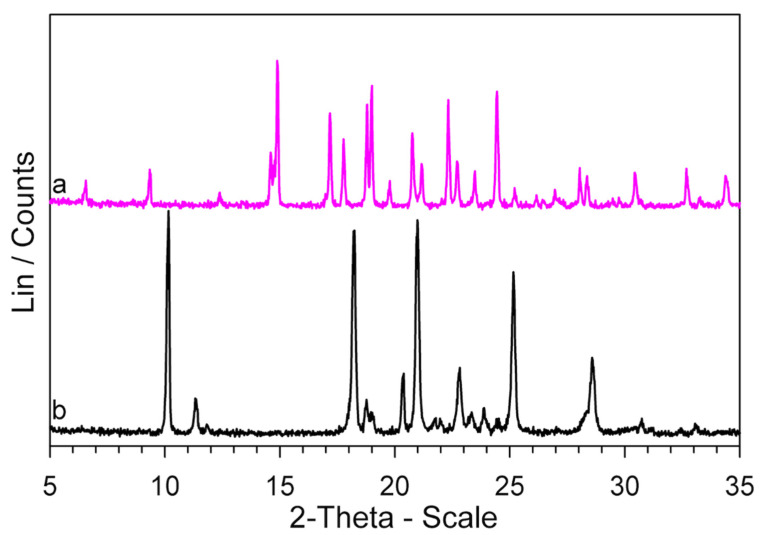
XRPD patterns of *GS* (**a**) and *GS** (**b**) samples.

**Figure 9 molecules-26-03142-f009:**
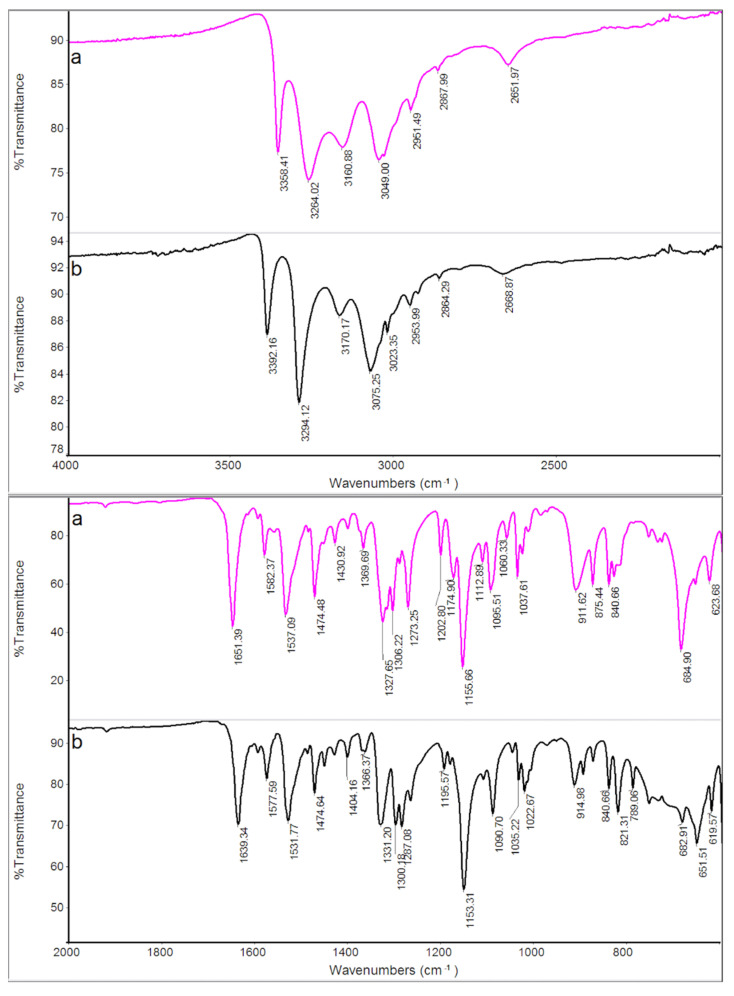
FT-IR spectra of *GS* (**a**) and *GS** (**b**) samples.

**Figure 10 molecules-26-03142-f010:**
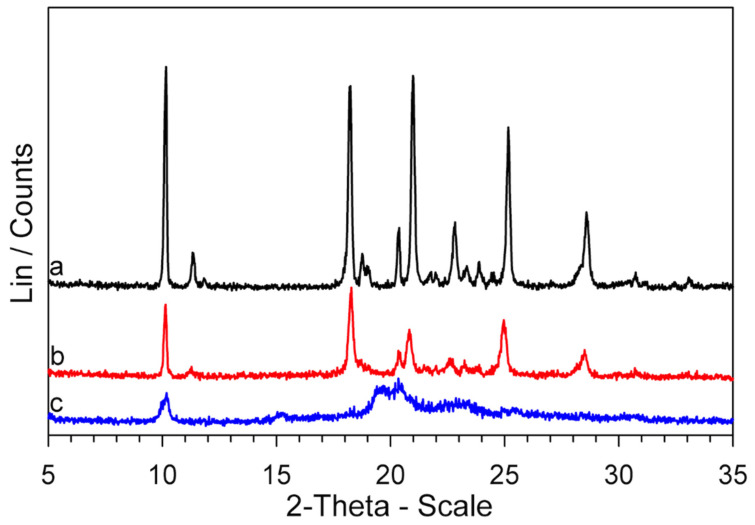
XRPD patterns of *GS** (**a**), $GPZ_{250}^{180}\;$(**b**), and *GPZ_250_* (**c**) samples.

**Figure 11 molecules-26-03142-f011:**
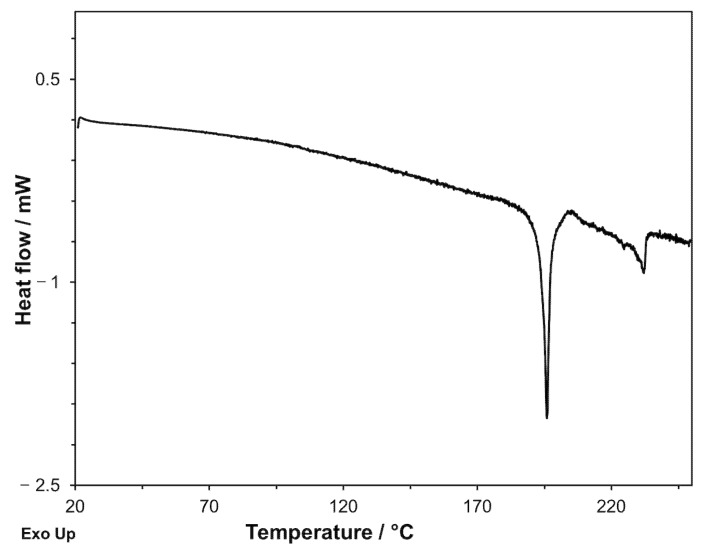
DSC curve registered on glipizide under nitrogen flow at a heating rate of 0.3 K·min^−1^.

**Figure 12 molecules-26-03142-f012:**
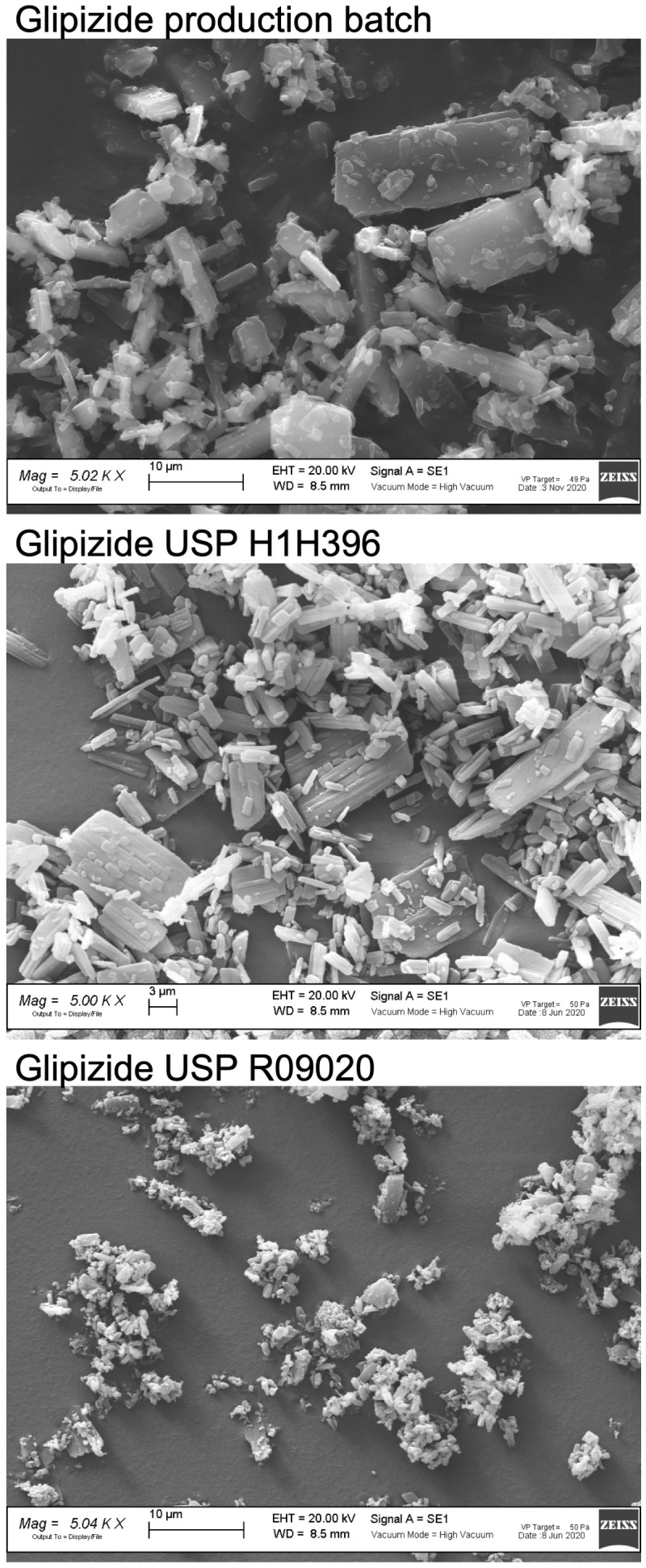
SEM images of a glipizide production batch, USP H1H396, and USP R09020 batches, 5000× magnification.

**Table 1 molecules-26-03142-t001:** Mass change, *T*_onset_, and *T*_f_ of the DSC peak measured by SDT on a *GPZ* production batch under nitrogen flow at different heating rates.

β, K·min^−1^	Δm%	*T*_onset_, °C	*T*_f_, °C
20	28.3	212.1	298.8
10	28.0	208.9	272.5
5	26.8	205.6	250.9
2	28.3	200.1	236.9
1	27.3	195.3	213.8
	Mean value ± s.d. 27.7% ± 0.7%		

**Table 2 molecules-26-03142-t002:** *T*_onset_, *T*_f_, and enthalpy change measured by DSC on a *GPZ* production batch under nitrogen flow at different heating rates.

β, K·min^−1^	*T*_onset_, °C	*T*_f_, °C	Δ*H*, J·g^−1^
30	217.7 ± 0.4	293.3 ± 4.6	291.2 ± 0.6
20	216.2 ± 0.4	285.1 ± 12.7	290.1 ± 0.5
10	214.5 ± 0.8	271.2 ± 3.5	290.8 ± 0.3
5	210.3 ± 0.3	257.3 ± 2.4	290.0 ± 0.3
2	205.2 ± 0.2	238.4 ± 1.4	290.2 ± 0.1
1	201.3 ± 0.4	229.2 ± 3.8	286.8 ± 3.7
0.5	196.1 ± 0.1	212.8 ± 5.1	292.2 ± 2.2
			Mean value ± s.d. 291.0 ± 0.6

**Table 3 molecules-26-03142-t003:** Temperature and time of the isothermal steps performed on one of the production batches of glipizide in TG furnace and codes of the recovered solid samples.

Temperature of Isothermal Step, °C	Time of Isothermal Step, min	Sample Code
170	210	$GPZ_{170\;{}^{\circ}C}^{210\;min}$
175	150	$GPZ_{175\;{}^{\circ}C}^{150\;min}$
180	135	$GPZ_{180\;{}^{\circ}C}^{135\;min}$
185	125	$GPZ_{185\;{}^{\circ}C}^{125\;min}$

**Table 4 molecules-26-03142-t004:** Main peaks/bands of FT-IR spectra of *GS* and *GS** (b) samples.

*GS* cm^−1^	*GS** cm^−1^
3358	3392
3264	3294
3160	3170
3049	3075
1651	1639
1582	1578
1537	1532
1328	1331
1306	1300
1273	1268
1156	1153
1095	1091
911	915
829	821

**Table 5 molecules-26-03142-t005:** Kinetic parameters, half-life and shelf-life obtained with Kissinger and Ozawa–Flynn–Wall methods.

Method	E_a_ kJ·mol^−1^	A min^−1^	*K* at 25 °C min^−1^	$t_{½}\mathrm{at}\;25\;{}^{\circ}C$ Years	t_90_ a 25 °C Years
Kissinger	364.2	1.8 × 10^39^	2.8 × 10^−25^	4.7 × 10^18^	7.2 × 10^17^
Ozawa–Flynn–Wall	353.9	1.3 × 10^38^	1.3 × 10^−24^	9.9 × 10^17^	1.5 × 10^17^

## Data Availability

Data available on request.

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
