# Peer review of "The Physico-Chemical Properties of Glipizide: New Findings"

_molecules, 2021, doi:10.3390/molecules26113142_

Round 1

Reviewer 1 Report

The submitted study focuses on the solid glipizide, an important API. From the introduction it is clear that the initial motivation, in my opinion, was to correct the work of Renuka et al. Due to the high quality of measurements and experience of the Authors the previously unknown phenomenon was discovered and described. The Authors have also “accidentally” found another polymorph of GS, great work!

 I believe in the explanation presented by the Authors and I like the narrative “story-like” style of this work.  The manuscript is surely well constructed and the performed analysis were justified, however there are some issues that need to be addressed before it can be proceeded further.

In the introduction the very important work (DOI: 10.1021/acs.cgd.6b01804 ) was not introduced. It presents the results of similar analysis on GPZ. I think the results from this work should be compared and discussed with the corresponding ones from your study. To make it clear, I am not an author of this publication.

Besides, the Authors should also present some information on the crystal structure of GPZ that has been deposited in the CCDC in 2017.

Figure 2S, the Authors should add a simulated PXRD, based on the CIF (i.e. refcode SAXFED01 ).

Lines 253-258 and Figures 3,4- I think that the molar masses of the “parts” of GPZ should be somehow introduced in the Figures. Is the molar mass of 5-methyl-2-pyrazinecarboxylic acid the same as the sum of cyclohexanamine and CO2? Is the molar mass of 5-methyl-N-[2-(4-sulphamoylphenyl)ethyl]pyrazine-2-carboxamide the same as 1-cyclohexyl-3-[[4-(2aminoethyl)phenyl]  sulfonyl]urea ?

Line 355, so how can you explain that FT-IR can be successfully used to distinguish the polymorphic forms of other solid organics?

Line 363, haven’t you tried to record the crystal structure of GS* using SCXRD? Since you already had a sample of it? Alternatively, you can use the already registered PXRD (i.e. those from Figure 8) to do some powder processing and at least determine the unit cell dimensions and space group of GS* via Pawley refinement. This can be done very quickly, this is a routine analysis and can be done using a lot of different software, both commercial and GPL.

Figure 8, could you please add theoretically simulated PXRD of GS? I guess its structure can be found in CCDC.

Reviewer 2 Report

This manuscript reports the physico-chemical studies of the thermal behavior of glipizide. With the use of different analytical techniques such as DSC, TGA, simultaneous DSC-TGA, XRPD, FT-IR and SEM, the authors analyzed the process responsible for the endothermic peak present in the DSC curves of glipizide, and showed that the DSC peak results from a decomposition process that involves gas evolution and formation of 5-methyl-N-[2-(4-sulphamoylphenyl) ethyl] pyrazine-2-carboxamide. Further analysis identified a new polymorph of 5-methyl-N-[2-(4-sul-phamoylphenyl) ethyl] pyrazine-2-carboxamide, and helped propose the reason why different batches of glipizide show the endothermic peak at different temperatures. At last, the authors performed kinetic studies of the decomposition process of glipizide based on the shift of the maximum temperature of the DSC peak with the heating rate, which suggested high stability of the solid glipizide at room temperature. In general, the experiments in this manuscript are clearly described and the conclusions are reasonably supported by the data presented. However, there are some issues that still need to be addressed:

  1. Why simultaneous DSC-TGA and DSC result in different Tonset when analyzing GPZ?
  2. Although FT-IR and XRPD suggest that the decomposition residue is 5-methyl-N-[2-(4-sulphamoylphenyl) ethyl] pyrazine-2-carboxamide, these two analytical methods are qualitative and thus not convincing enough. Please use HPLC-MS to further confirm the identity of the decomposition product.
  3. Does the lower Tonset of GPZ3h also result from smaller particle size? As it was reported in literature that the amorphous form of GPZ can be obtained by grinding, did authors observe amorphous form in other GPZ samples, for example GPZ5h and GPZ8h?
  4. Does GS decompose during DSC analysis?
  5. Some of the abbreviations are defined in the abstract or Material and methods section, and this makes the manuscript hard to read. Please consider defining the abbreviations in the main text.

Reviewer 3 Report

The manuscript “The Physico-Chemical Properties of Glipizide: New Findings”, by Giovanna Bruni, Ines Ghione, Vittorio Berbenni, Andrea Cardini, Doretta Capsoni, Alessandro Girella, Chiara Milanese and Amedeo Marini, presents interesting information on glipizide and its physical-chemical properties. Since glipizide is a molecule having biological activity applicable to therapeutical purposes, the study of its properties should be conducted as deep as possible. In this work, the authors provide complementary data to previous studies and, also, they contribute with a critical view to earlier measurements (and interpretations). I consider this an important contribution to the scientific discussion.

In my opinion, the Abstract should be a single paragraph. With some exceptions, this is the normal practice.

The introduction provides the background of this work and the motivation. Also, the aim of the work was indicated in this section. The text has some mistakes. I included some comments in the PDF document (attached).

The description of materials and methods is sufficient. Details on the compound, grounding and analytical techniques are provided.

The quality of some figures is high enough to distinguish the details. The quality of other figures can be improved. Data of the DSC-TGA curves, FT-IR spectra and XRPD patterns can be saved as ASCII files. In this way, the authors can plot the data with software like Origin and then export it to an image format with a good resolution. The width of the curves can easily be modified to make them visible even if the size of the image is reduced. Also, the size of numbers and letters can easily be changed. Please improve the quality of (at least) Figures 5, 6, 8, 9 and 10.

In the Supplementary Materials, I suggest figure labels with the format "Fig. S1", "Fig. S2" and so on, instead of "Fig. 1S", "Fig. 2S" and so on. Also, in the manuscript the authors write “Figure”, and “Fig.” in the Supplementary Material. This must be uniform.

There are 28 references, half of them are of the last 10 years. It is acceptable (in my opinion), however, it would be better if the authors add a couple of recent references. I think that these references could be added in the introduction to highlight the current interest of this work.

In summary, I conclude that this paper requires some corrections, none of them are critical. I believe that the authors explained well what they try to present to the readers. After minor revisions, this paper could be considered for publication in Molecules.

Round 2

Reviewer 1 Report

The Authors have answered my comments and made the corrections. The manuscript can now be published as it is.